# Split-Attention Networks with Self-Calibrated Convolution for Moon Impact Crater Detection from Multi-Source Data

Yutong Jia [1] , Gang Wan [1,*], Lei Liu [1], Jue Wang [1], Yitian Wu [1] , Naiyang Xue [2], Ying Wang [1] and Rixin Yang [1]

[1] Department of Surveying and Mapping and Space Environment, Space Engineering University, Beijing 101407, China; jiayutong@st.btbu.edu.cn (Y.J.); liuleiargis@pku.edu.cn (L.L.); 3120160445@bit.edu.cn (J.W.); wuyt@radi.ac.cn (Y.W.); wang18922950496@163.com (Y.W.); naruto_young@163.com (R.Y.)

[2] Department of Electronic and Optical Engineering, Space Engineering University, Beijing 101407, China; hgdxny15@163.com

\* Correspondence: casper_51@163.com; Tel.: +86-131-4521-4654

**Abstract:** Impact craters are the most prominent features on the surface of the Moon, Mars, and Mercury. They play an essential role in constructing lunar bases, the dating of Mars and Mercury, and the surface exploration of other celestial bodies. The traditional crater detection algorithms (CDA) are mainly based on manual interpretation which is combined with classical image processing techniques. The traditional CDAs are, however, inefficient for detecting smaller or overlapped impact craters. In this paper, we propose a Split-Attention Networks with Self-Calibrated Convolution (SCNeSt) architecture, in which the channel-wise attention with multi-path representation and self-calibrated convolutions can generate more prosperous and more discriminative feature representations. The algorithm first extracts the crater feature model under the well-known target detection R-FCN network framework. The trained models are then applied to detecting the impact craters on Mercury and Mars using the transfer learning method. In the lunar impact crater detection experiment, we managed to extract a total of 157,389 impact craters with diameters between 0.6 and 860 km. Our proposed model outperforms the ResNet, ResNeXt, ScNet, and ResNeSt models in terms of recall rate and accuracy is more efficient than that other residual network models. Without training for Mars and Mercury remote sensing data, our model can also identify craters of different scales and demonstrates outstanding robustness and transferability.

**Keywords:** crater detection algorithm (CDA); R-FCN; self-calibrated convolution; split attention mechanism; transfer learning; remote sensing

## 1. Introduction

Impact craters are considered to be one of the most important features of the Moon, Mars, and Mercury [1]. They gradually evolve because of colliding objects, such as meteorites, satellites, or massive asteroids [2]. Most of the impact craters on the lunar surface have circular pit structures with different sizes and uneven aggregations.

The impact craters on the surface of deep space stars contain significant geological data. This is because they are the product of the meteorite's high-speed movement, impact on the surface of celestial bodies, and lava eruption inside heavenly bodies. Therefore, such data can be used to retrieve the geological age of the stars [3], analyze the tectonic history of the lead [4], and explore the existence of iced water [5]. In addition, it can be used for autonomous navigation [6], landing site selection [7], base selection, and other missions of deep space probs.

The precise and rapid discovery of impact craters has always been a priority for deep space exploration since the beginning of the Moon and Mars exploration activities. Several deep space star surface impact crater extraction algorithms have also been proposed. These algorithms are broadly classified as (i) traditional algorithms, which use image processing

technology to identify impact craters, and (ii) automatic algorithms [8–11], which use deep learning models to extract impact craters [12–14].

The traditional automatic feature extraction algorithms for impact crater morphology are mainly based on classical image processing methods, including Hough transform, feature matching, curve fitting, and other recognition techniques. For example, [15] used the Hough Transform to obtain more than 75 percent of the current impact craters with a diameter greater than 10 km based on data from the Mars Orbiter Laser Altimeter (MOLA). Hough transform is the most widely used method in this area which is efficient for impact crater identification and recognition of the discontinuous edges. However, for irregular shapes, the computational complexity of such methods is very high. Further, [16] used the conic curve-fitting approach to automatically classify asteroid impact craters to aid optical navigation of the spacecraft to solve this problem. The proposed method in [15] successfully identified about 90% of impact craters with an error rate of less than 5%. Based on the Mars Orbiter Camera (MOC), Mars Orbiter Laser Altimeter (MOLA), and High-Resolution 3D Camera (HRSC)), [9] proposed a least-squares fitting method (DLS) for the identification of Mars impact craters. By comparing the recognition results of the Hough ring transform algorithm, they then showed that the conic fitting method is more reliable, but its computational complexity is higher.

The construction and matching of data quality and crater characteristics are central to traditional crater recognition algorithms. The main goals are to create a more accurate crater function model and a faster template matching algorithm. Nonetheless, the geomorphic features of impact craters are many. The impact craters in an area may also be nested and overlapped. The available data samples are also insufficient in many cases.

Artificial intelligence has developed rapidly by introducing deep learning models in recent years. Among deep learning techniques, convolutional neural networks (CNN) are shown to offer significant practical advantages for image processing. CNN have been successfully applied to many classic image processing problems, such as image denoising, super-resolution image reconstruction, image segmentation, target detection, and object classification. Crater detection and segmentation of the image data can be used to solve the problem of crater recognition.

Cohen [17] considered the classification of meteorite craters, proposing a meteorite crater identification and classification algorithm based on a genetic algorithm. Yang [3] also proposed an impact crater detection model on the lunar surface based on the target detection R-FCN model and further studied the lunar age estimation. Furthermore, [12] suggested the DeepMoon model for lunar surface impact crater identification based on the U-Net model of image semantic segmentation in deep learning. They then transferred their model to the Mercury surface impact crater recognition and achieved reasonable results. The DeepMoon model's structure was applied to the impact craters on Mars' surface in [18], and the DeepMars model was proposed to achieve rapid detection of impact craters on Mars' surface. Jia [19] also improved the model and suggested a need-attention-aware U-NET (NAU-NET) in the DEM impact crater trial and obtained Recall and Precision of 0.791 and 0.856, respectively.

Intelligent impact crater identification methods based on deep learning are more efficient than the traditional identification methods in recognizing significant differences in the radius of the impact crater and their complex morphological characteristics. However, due to the variety of deep space objects, the recognition model based on single star surface impact craters offers a poor generalization ability, especially in recognizing overlapping and small impact craters. To address this issue, in this paper, we consider the deep space star surface impact crater and combine the existing Moon image and DEM data of the Moon, Mars, and Mercury surfaces to establish a deep learning-based deep space star surface impact crater intelligent identification framework. The proposed model improves the model generalization ability through transfer learning. An improved residual network and multi-scale target extraction are introduced to accelerate the model convergence and improve the accuracy of feature extraction. In addition, a more efficient pooling operation

and Soft-NMS algorithm are proposed, which effectively reduces false-negative errors of the detection model.

The main contributions of this paper are as follows:

1.  We propose a SCNeSt architecture in which the channel-wise attention with multi-path representation and self-calibrated convolutions provide a higher detection and estimation accuracy for small impact craters.
2.  To address the issues caused by a single data source with low resolution and insufficient impact crater features, we extract the profile and curvature of the impact crater from Chang 'e-1 DEM data, integrated it with Chang 'e-1 DOM data, and combined it with International Astronomical Union (IAU) impact crater database, and constructed the VOC data set.
3.  The lunar crater model is trained, and transfer learning is used to detect the impact craters on Mercury and Mars. This is shown to increase the model's generalization ability.

The rest of this paper is organized as follows. In Section 2, we introduce the R-FCN network for target detection and SCNeSt, RPN, and ROI Pooling. The model is then applied for impact crater detection on Mercury and Mars surfaces using transfer learning. Section 3 then introduces the experimental data, evaluation indexes, and experimental conditions. Furthermore, Section 4 evaluates the lunar impact crater detection results and compares the proposed network with other existing networks. Finally, Section 5 provides our conclusions and offers insights on the direction of future work.

## 2. Methods

We adopted a combination of deep learning and transfer learning, as shown in Figure 1. In the first stage, CE-1 images of $4800 \times 4800$ pixels and $1200 \times 1200$ pixels were used (image fusion method referred to 3.1), achieving a recall rate of 95.82%, where almost all identified craters in the test set were recovered. In the second stage, we transferred the detection model of the first stage to the SLDEM [20] images without any training samples. The learning process in the second stage followed transfer learning, hence extracts the learning features and knowledge from the SLDEM data with a recall rate of 91.35%. We finally found 157,389 impact craters on the Moon, ranging in size from 0.6 to 860 km. The number of detected craters was almost 20 times larger than the known craters, with 91.14 percent of them smaller than 10 km in diameter.

For the meteorite craters that were in both CE-1 and SLDEM, we selected $D \geq 20$ km for CE-1 detection, and $D < 20$ km for SLDEM data detection. The average detection time of an image was 0.13 s.

### 2.1. SCNeSt Backbone Network

Inspired by the ResNeSt network framework and the self-calibrated convolution in the ScNet [21], in this paper, we improved the ResNeSt. To enhance the diversity of output features, self-calibrated convolution in the ScNet was substituted with the second convolution layer of the ResNeSt Block to obtain more features and more efficient classification performance. Meanwhile, in a split-attention radix group of ResNeSt, we used the method of combining MaxPooling and AvgPooling to replace the original GlobalPooling. This enabled obtaining more texture features at the same time. MaxPooling reduces useless information, and AvgPooling obtains the texture information.

The SCNeSt Block structure is shown in Figure 2. The self-calibrated Conv evenly divided the input into four parts and then performed different operations for each position. First, the input $X$ was evenly divided into and various functions that process the input $X$. Then, $X_1$ was sent up to the first branch (self-calibrated branch) and $X_2$ to the second branch (conventional transform branch). Finally, the processed features were concatenated as the output.

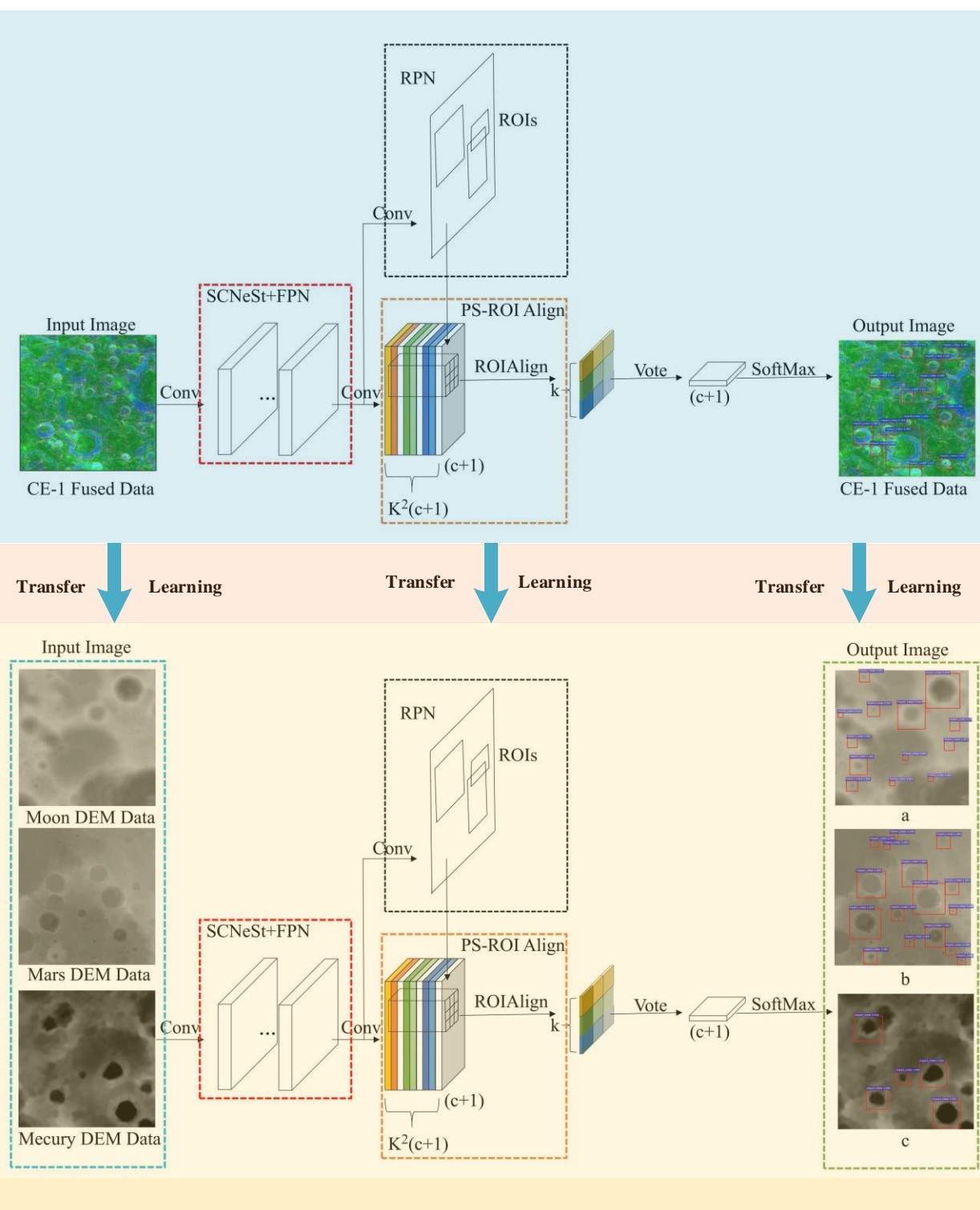

**Figure 1.** Deep space impact crater detection framework based on the improved R-FCN.

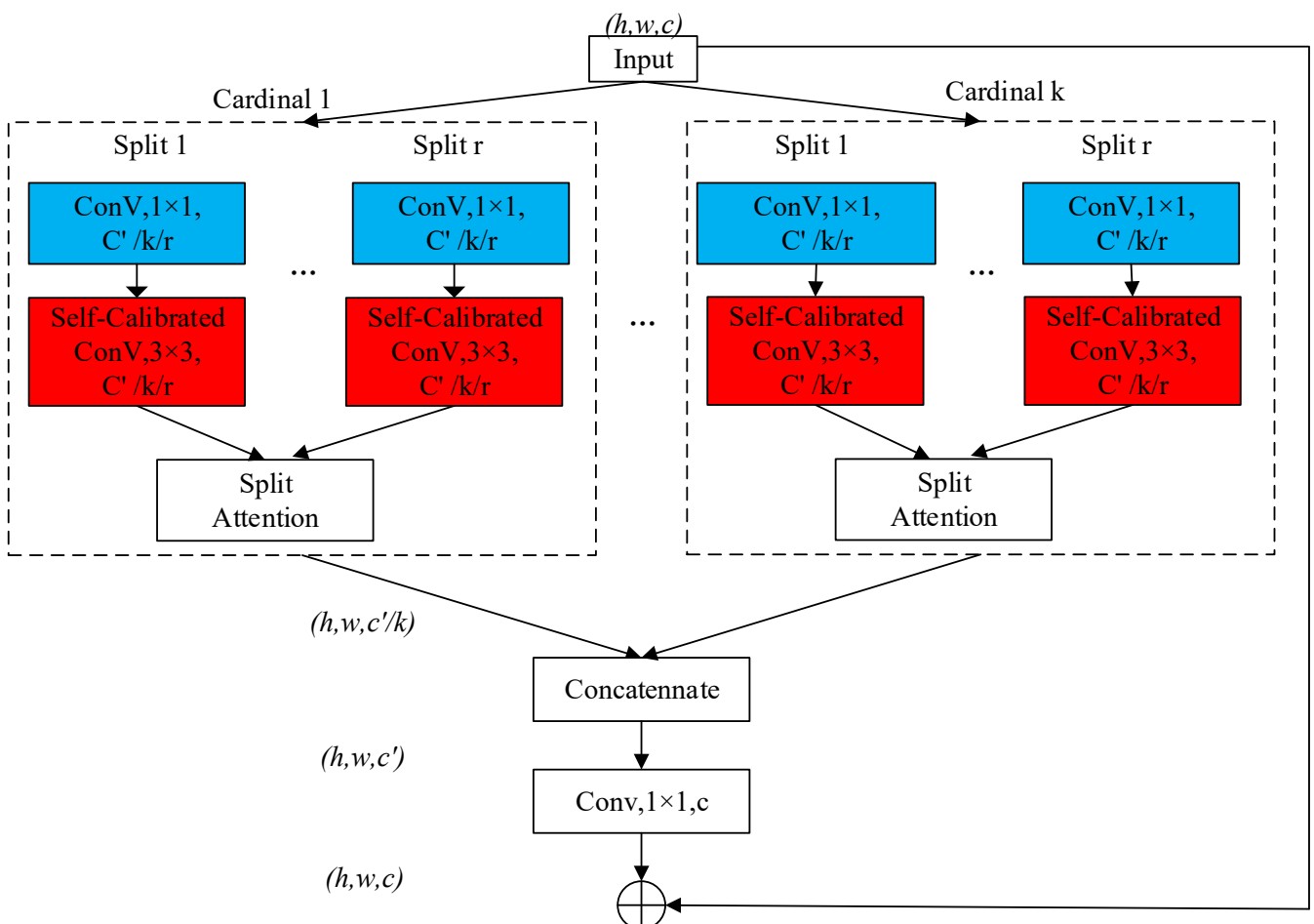

**Figure 2.** The SCNeSt block. The blue module represents vanilla convolutions, and the red module describes self-calibrated convolutions.

In the self-calibrated branch, for input $X_1$, average subsampling, convolution feature transformation, and bilinear up-sampling were performed. The input was then added to obtain the attention feature map at the spatial level. The acquired spatial attention map was fused with the transformed $X_1$. The process is described as:

$$\begin{cases} X'_1 = Up(T_1) = Up(T_1 \times K_2) = Up(Down(X_1) \times K_2) \\ Y'_1 = F_3(X_1) + \sigma(X_1 + X'_1) \end{cases} \tag{1}$$

The schematic diagram of the self-calibrated Conv module is shown in Figure 3. The self-calibrated Conv proposed in this paper has the following three advantages:

(1) Self-calibrated branching significantly increases the receptive field of the output features and acquires more features.

(2) The self-calibrated branch only considers the information of the airspace position, avoiding the information of the unwanted region, hence uses resources more efficiently.

(3) Self-calibrated branching also encodes multi-scale feature information and further enriches the feature content.

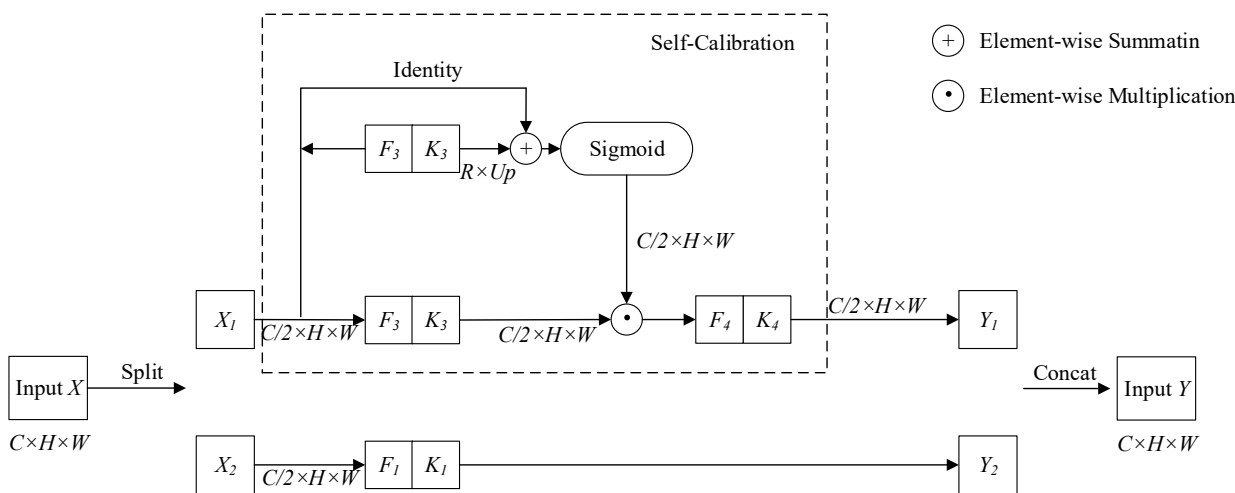

**Figure 3.** The schematic diagram of the self-calibrated Conv module. In self-calibrated convolutions, the original filters were separated into four portions, each in charge of different functionality. This makes self-calibrated convolutions quite different from traditional convolutions or grouped convolutions performed homogeneously.

### 2.2. Multi-Scale Feature Extractor

Although the external network detects small targets, the external network has weak semantics. If we only carried out the deconvolution operation without feature fusion, part of the information would be lost after repeated convolution and deconvolution. This is more harmful to detecting the small targets. To address this issue, we synchronized with the deconvolution process, and the high-level features were successively fused with the shallow elements. This preserved the semantic information and resolution of the feature layer.

The FPN [22] consisted of three parts, as shown in Figure 4d. The first part was the feature extraction using the feedforward process of the general convolutional neural network from bottom to top.

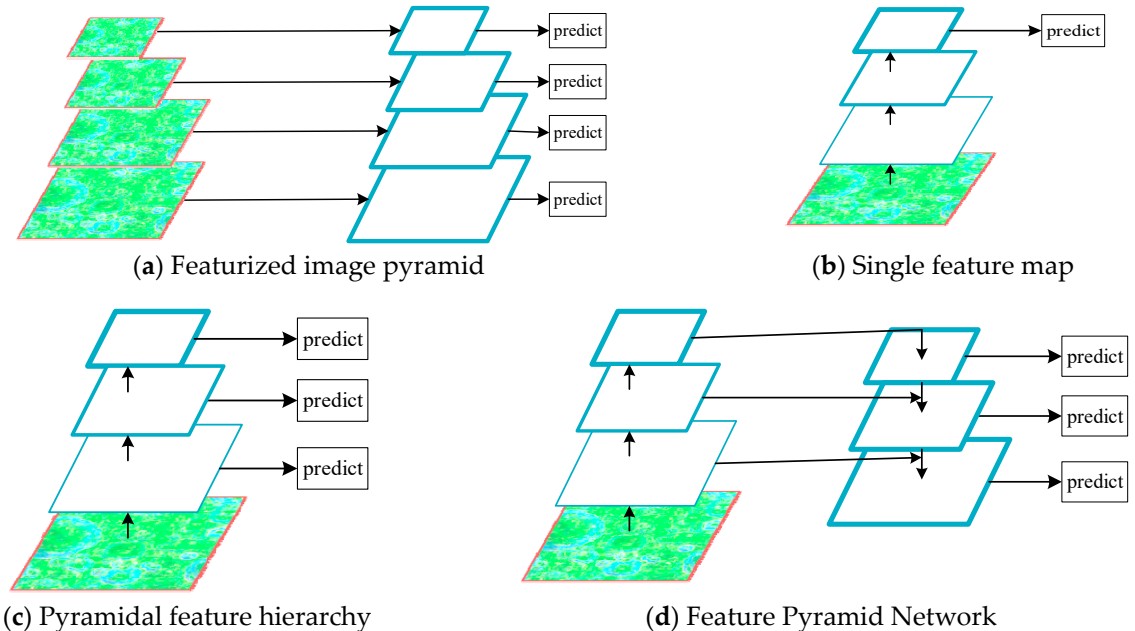

**Figure 4.** The multi-scale detection methods.

In the second part, we first selected the upper-level feature graphs with more vital semantic information in the feature graphs obtained in the first part. Then, they were up-sampled from top to bottom to strengthen the upper-level features. This also equalized the sizes of the feature graphs in the adjacent layers. In the third part, the feature graphs of the first two steps were combined using horizontal connections. Through these three parts, the high- and low-level features were connected to enrich the semantic information of each scale.

The whole FPN network was embedded into the RPN to generate features of different scales. These features were then fused as the input of the RPN network to improve the accuracy of the two-stage target detection algorithm, as shown in Figure 5.

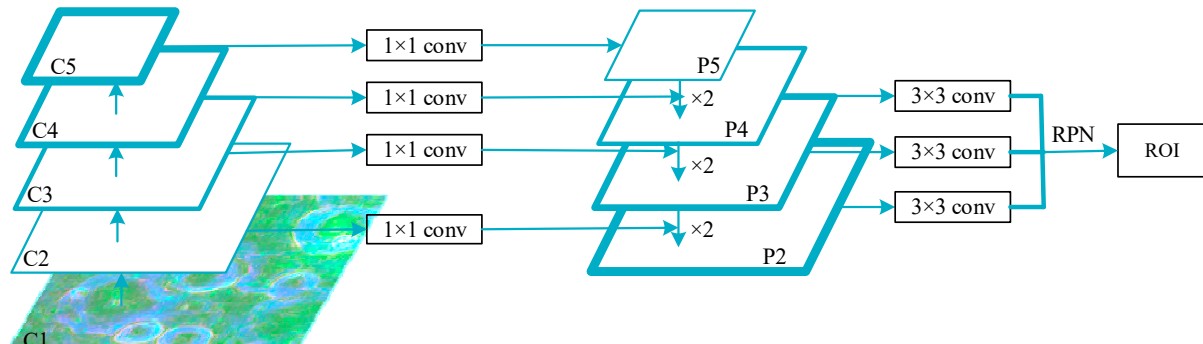

**Figure 5.** RPN network with FPN.

### 2.3. Position-Sensitive ROI Align

The ROI Pooling layer [23] improves the detection accuracy and speeds up the training and testing process. Nevertheless, two rounds of quantization operations were required, generating the candidate box and determining the corresponding grid position. The first step was to round up the two sampling points selected in the original ROI Pooling layer. This ensured that the generated sampling points were aligned with the standard coordinate points, and the subsequent Pooling operations would round up again. Since the feature map obtained by the CNN was 16 times smaller than that of the original image, X/16 needed to be used for the calculation in the corresponding process. Hence, there existed floating-point numbers with decimals in the calculations. The coordinate point deviation on the feature map caused by the two-step rounding operation corresponded to the pixel deviation on the original image, which was 16 times. The pixel deviation led to mismatching between the image and the feature map so that the ROI on the feature map could not correspond to the original image. This, however, had an impact on the regression positioning of the back layer.

To avoid the round-off operation of the floating-point numbers by two rounds quantization, a bilinear difference pair was introduced to improve the alignment method. A particular region of the feature map corresponding to the ROI was divided into $2 \times 2$ region blocks. Each region block was then quartered, and each small grid center was taken as the sampling point. As illustrated in Figure 6, the coordinates of the 16 sampling points in vertices A, B, C, D, and the evenly divided $2 \times 2$ region were not integers. After determining the sampling points, the bilinear difference evaluation was directly mapped to the feature map, and each sampling point was evaluated in the X and Y directions. After the difference was completed, the maximum pooling operation was carried out, and the final feature map was obtained by analogy. The whole procedure did not operate on specific coordinate values. The decimal was retained in the coordinate calculation process to avoid the discrete quantization error of the two ROI round-off operations and make the final detection box position more accurate.

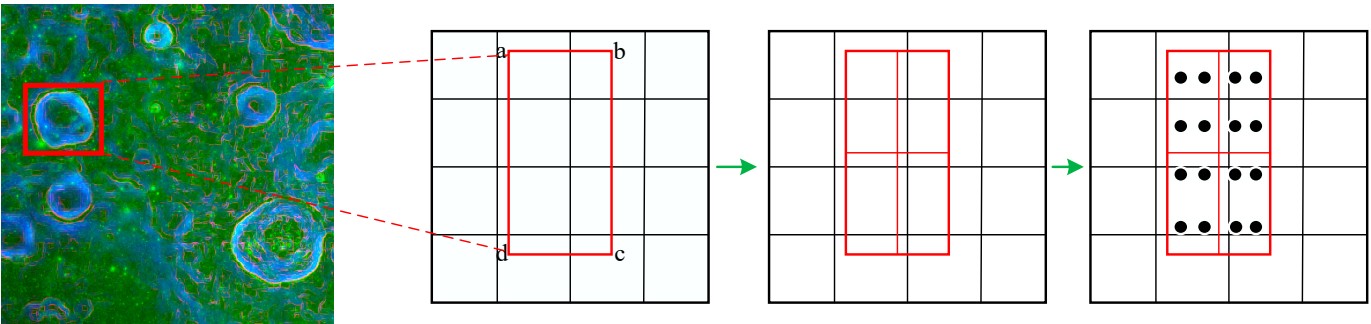

(**a**) RoI coordinates     (**b**) Coordinate mapping regression     (**c**) RoI Pooling     (**d**) Sampling point evaluation

**Figure 6.** The improved RoI Pooling using bilinear interpolation.

A Position Sensitive ROI Align algorithm was implemented by porting ROI Align into PS-ROI Pooling. The PS-ROI Align improved the detection performance of the model and significantly improved the perception ability for the small objects.

### 2.4. Soft-NMS

After obtaining the detection box by the R-FCN model, we used the non-maximum suppression (NMS) [24] algorithm to accurately convey the best coordinates of the target and remove the repeated boundary box. For the same object, multiple detection scores were generated as the detection windows were overlapped. In such cases, the NMS kept the correct detection box (with the highest confidence). The remaining detection boxes were removed from the optimal position (with the confidence reduced to 0) to obtain the most accurate bounding box. The NMS can be expressed by the score reset function:

$$Q_i = \begin{cases} Q_i, \ iou(M, b_i) < N_t \\ 0, \ iou(M, b_i) \geq N_t \end{cases} \tag{2}$$

where $Q_i$ is the confidence of the detection box, $M$ is the position of the detection box with the highest confidence, $b_i$ is the position of the detection box, $N_t$ is the set overlap threshold, and $iou(M, b_i)$ is the overlap rate between $M$ and $b_i$.

Note that non-maximum suppression may cause a critical issue by forcing the scores of adjacent detection boxes to 0. In such cases, if different impact craters appear in the overlapping area, the detection of impact craters will fail. This reduces the detection rate of the algorithm, as in Figure 7a.

Soft non-maximum suppression algorithm (Soft-NMS) [25] replaces the score reset in the NMS algorithm with:

$$Q_i \leftarrow Q_i f(iou(M, b_i)) \tag{3}$$

Noting that the impact craters were rectangular targets in the image, and considering overlapping impact craters, a linear weighted fraction resetting function was used as the following:

$$Q_i = \begin{cases} Q_i, \ iou(M, b_i) < N_t \\ Q_i(1 - iou(M, b_i)), \ iou(M, b_i) \geq N_t \end{cases} \tag{4}$$

In Figure 7b, the confidence of the dashed line detection box was changed to 1.0, indicating that Soft-NMS can effectively avoid missing the impact craters in the overlapping areas. This significantly improved the detection rate of the model.

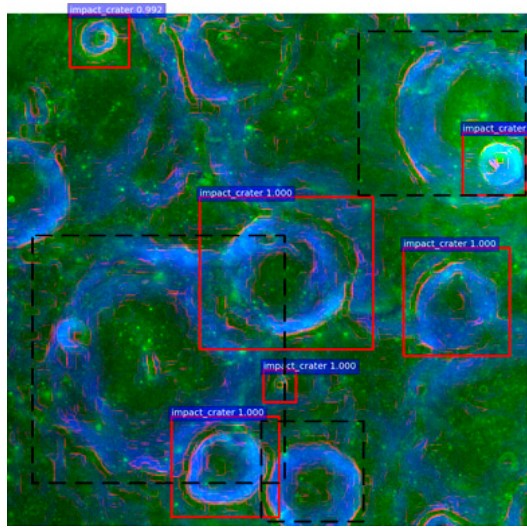 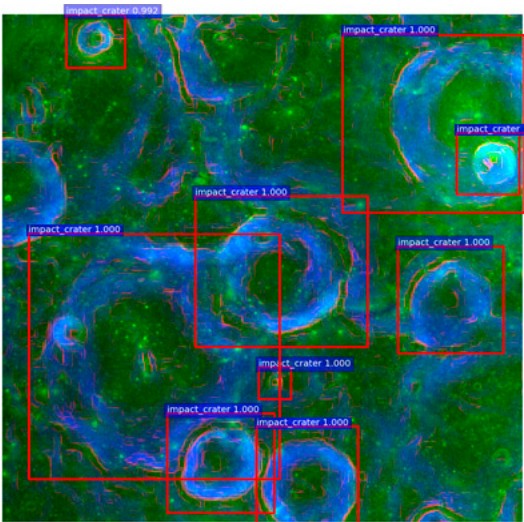

(**a**) Non-maximum suppression algorithm.                (**b**) Soft non-maximum suppression algorithm.

**Figure 7.** Comparison of NMS and Soft-NMS algorithms.

## 3. Experiments

Our algorithm was divided into two parts. First, the features of impact craters were extracted under the Structure of the R-FCN network based on the SCNeSt network skeleton, and the data were DOM and DEM fusion data from CE-1. Multi-scale Feature Extractor and Position-Sensitive ROI Align could better detect impact craters of different scales. They were combined with the Soft-NMS algorithm to accurately convey the best coordinates of the target and remove the repeated boundary box. In the first stage, the craters with D > 20 km were mainly extracted. In the second stage, the trained model was applied to SLDEM data to extract small craters with D < 20 km. What is more, the trained models were then applied to detecting the impact craters on Mercury and Mars using the transfer learning method.

### 3.1. Dataset

The area studied on the Moon was latitude $-65°{\sim}65°$, longitude $-180°{\sim}65°$, and longitude $65°{\sim}180°$. The DOM and DEM data adopt equiangular cylindrical projection. During the crater exploration mission, DEM data from CE-1 was resampled to 120 m/pixel. The slop information and profile curvature were also extracted from DEM data. DOM data was integrated with DEM data. The crater in the study area was marked by using the lunar data set published by the IAU impact crater VOC dataset generated by combining with Labelimg. The CE-1 fusion data were then clipped into $1200 \times 1200$, $4800 \times 4800$ images at a 50% overlap rate, 8000, 1000, and 1000 images were randomly selected and used for training, validation, and testing, respectively. Due to the low resolution of CE-1 data, we used it to identify large impact craters ranging from 20 km to 550 km in diameter. The detailed data generation was shown in Figure 8.

The SLDEM from the Lunar Reconnaissance Orbiter (LRO) and the Kaguya merged digital elevation model had a resolution of 59 m/pixel and spans ±60 degrees latitude (and the maximum range in longitude). The Plate Carree projection was used to create this global grayscale map, which had a resolution of $184,320 \times 61,440$ pixels and a bit depth of 16 bits per pixel. We cropped it into $1000 \times 1000$-pixel images to detect small impact craters. The SLDEM data has a high resolution and has a good identification effect for small impact craters and degraded impact craters. We used it to identify impact craters with a diameter less than 20 km.

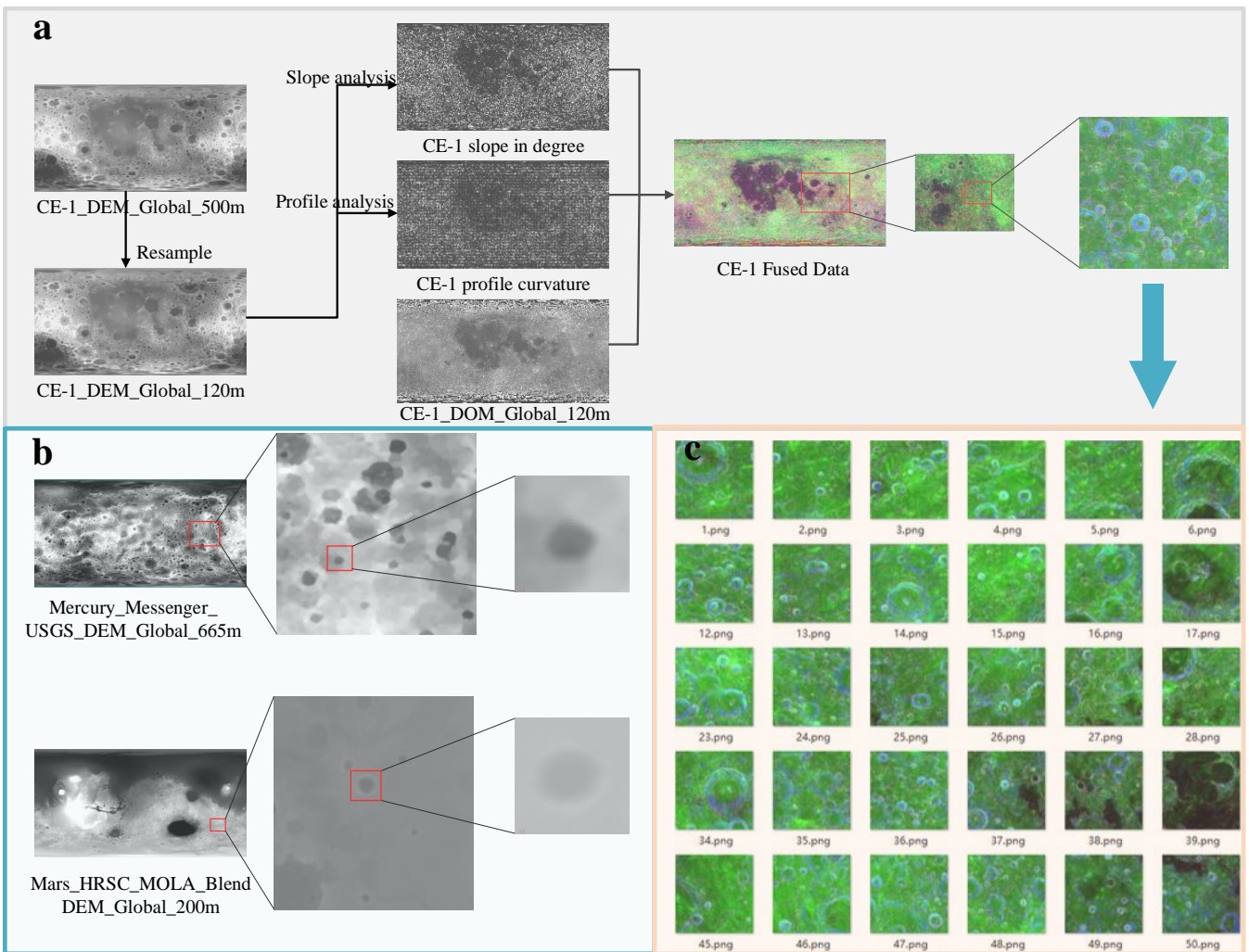

**Figure 8.** Deep space impact craters data: ((**a**) CE-1 data fusion process. (**b**). Mercury and Mars DEM data. (**c**). The CE-1 fusion dataset).

The Mercury MESSENGER Global DEM has a resolution of 665 m per pixel and spans ±90 degrees latitude and Longitude range from 0° to 360°, which is different from our Moon DEM in terms of image properties. This global grayscale map is an Equirectangular projection with a resolution of 23,040 × 11,520 pixels. Mercury differs from the Moon in gravitational acceleration, surface structure, terrain, and impact background.

The Mars HRSC and MOLA Blended Global DEM had a resolution of 200 m per pixel and spans ±90 degrees latitude (and the maximum range in longitude). This global grayscale map was a Simple Cylindrical projection with a resolution of 106,694 × 53,347 pixels. We also cropped it into 1000 × 1000-pixel images to detect small impact craters.

### 3.2. Evaluation Metrics

Computer configuration in the experiment comprised two NVIDIA GeForce 2080 Ti RTX GPUs, 64 Gb of memory, Ubuntu16.04 operating system, Cuda10.0, Cudnn7.5, and Opencv3.5.6, and used Caffe framework for training.

The Precision–Recall (P-R) curve and Average Precision (AP) values were used in this experiment to objectively test the accuracy of the target detection algorithm.

$$P = \frac{N_{tp}}{N_{tp} + N_{fp}} \tag{5}$$

where $N_{tp}$ is the number of correctly detected crater targets in the formula, and $N_{fp}$ is the number of miss-detected targets. The Recall in the P-R curve represents the missed detection rate of the algorithm:

$$R = \frac{N_{tp}}{N_{tp} + N_{fn}} \tag{6}$$

where $N_{fn}$ is the missed meteorite crater target.

With Precision as the longitudinal axis and Recall as the horizontal axis, the P-R curve was then fitted by changing the threshold condition. In addition, for the target detection task, the IOU of the predicted location and the actual location of the target were considered when calculating the P-R curve. This was to reflect the accuracy of the target location prediction. In this experiment, IOU was set to 0.5.

The $F_1$ value is a statistical index used to measure the accuracy of the dichotomous model. This index takes into account both the accuracy and recall rate of the classification model. The $F_1$ value can be defined as a weighted average of model accuracy and recall rate as:

$$F_1 = 2 * \frac{PR}{P + R} \tag{7}$$

where $P$ and $R$ are the accuracy and recall rates, respectively.

### 3.3. Training Details

In training the convolutional neural network, it is necessary to set some super parameters, e.g., learning rate, training iteration volume, selection of loss function. The parameter settings are shown in Table 1.

**Table 1.** The model super parameters.

| Parameter | Value |
| --- | --- |
| Learning rate | 0.0001 |
| Training batches | 10,000 |
| Training wheels | 1000 |
| Objective function | Cross-entropy and MSE |

We used the Adam algorithm for optimization with the momentum of the SGD gradient descent algorithm. We used the first-moment estimation and second-order moments of the gradient vector to estimate the dynamic adjustment of each parameter. In each iteration update, the iteration vector had a specific scope to stabilize the parameter. The introduction of the near iterative gradient direction of the penalty term improved the convergence speed of the models.

The objective function was divided into classification and regression. The Mean Square Error (MSE) algorithm realized the target location by calculating the lowest square value of the predicted site and the actual location. The cross-entropy function also calculated the probability difference between the prediction confidence of the target classification and the essential target category. Furthermore, having the cross-entropy as the loss function prevented the learning rate reduction in the MSE loss function in the case of gradient descent. Therefore, we set

$$C = -\frac{1}{N} \sum_n y \ln a + (1 - y) \ln(1 - a) \tag{8}$$

to be optimized where $y$ is the expected output, $a$ denotes the actual output, $N$ is the total number of training data, $n$ represents the input sample.

## 4. Results and Discussion

### 4.1. Analysis of the Lunar Impact Crater Detection Results

In Figure 9, we compare the proposed model in this paper with the identified crater distribution. As it is seen, the number of identified lunar craters was significantly higher than that of the number of identified craters with diameters between 1 and 100 km. This indicates that the proposed model identified many craters in the small and medium diameter ranges. Despite the irregular, severely eroded, and scattered nature of the major lunar craters, the proposed model recognized 46 craters with diameters ranging from 200 to 550 km.

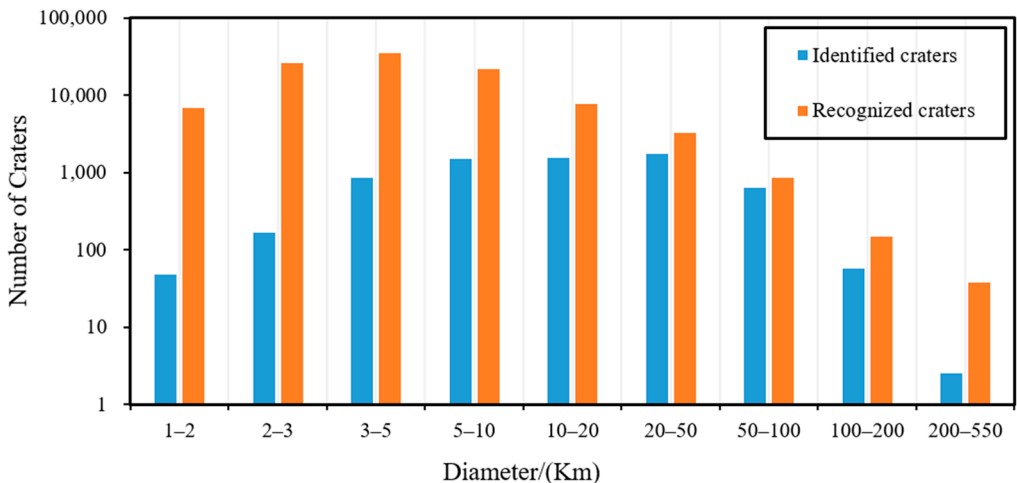

**Figure 9.** Comparison of the distribution of lunar craters with different diameters identified by the IAU. (The yellow column represents the number of craters recognized by the model. The blue column represents the number of identified craters.).

We also studied the detected craters to ensure their authenticity. We compared them to three databases of artificially acquired lunar craters:

(1)　Head et al. [26], where a total of 5185 craters with a diameter of D ≥ 20 km was obtained by the Digital Terrestrial Model (DTM) of the Lunar Reconnaissance Orbiter (LRO) Lunar Orbiter Laser Altimeter (LOLA);

(2)　Povilaitis et al. [27], in which the previously described database was expanded to 22,746 craters with D = 5–20 km;

(3)　The Robbins database [28] holds over 2 million lunar craters, including 1.3 million with D ≥ 1 km. This database contains the largest number of lunar craters.

In addition, three kinds of automatic crater directories were considered:

(4)　Salamunićcar et al. [29], in which LU78287GT was generated based on Hough transform;

(5)　Wang et al. [30], which was based on CE-1 data, and included 106,016 impact craters with D > 500 m;

(6)　Silburt et al. [12], which was based on the DEM data from CNN and LRO and generated a meteorite crater database.

(7)　Yang et al. [3] adopted the CE-1 and CE-2 data and compiled 117,240 impact craters with D ≥ 1–2 km.

Figure 10 shows the comparison results of the number of matched craters at different scales. For manual annotation, it is seen that the matching degree of Povilaitis et al. is consistent with that obtained in our model for craters with diameters of 5–550 km. For the manually annotated Robins database, the number of craters between 1 and 2 km is close to the number identified by our model. This is because of the efficiency of the proposed model in the identification of smaller craters. However, the number of craters between 2 and 20 km is far greater than that of our model. This is because degradation of craters and other reasons leads to insufficient feature extraction. For the overall matching percentage

of manually annotated data, the consistency of our recognition results reaches 88.78% for craters with diameters between 5–550 km.

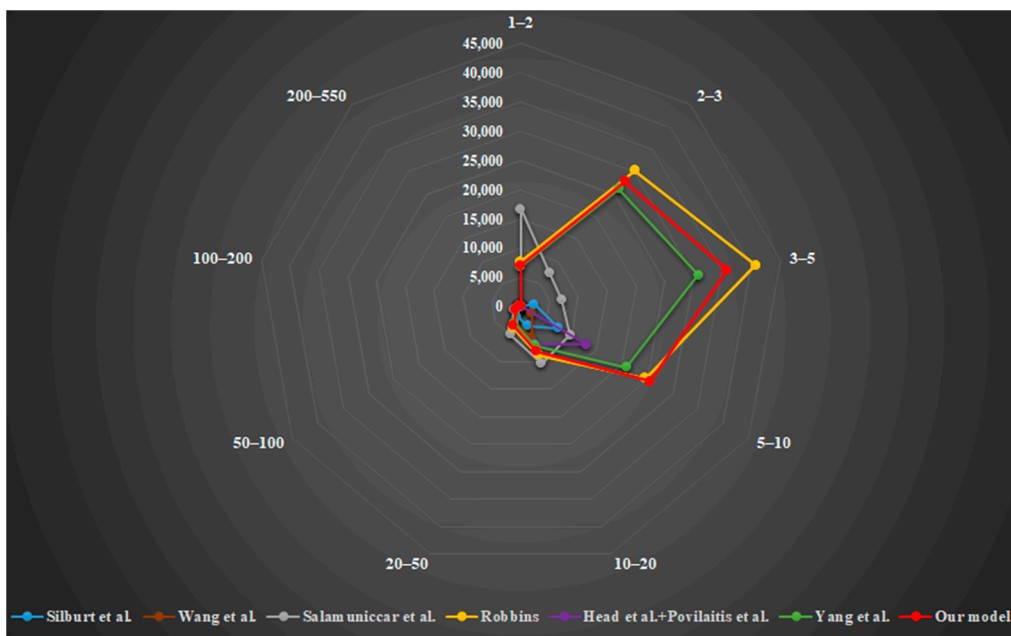

**Figure 10.** Comparison results of the number of the matched craters at different scales.

For the automatically labeled database and Yang's database, and the impact craters diameter D ranging from 1 to 5 km, our model outperformed the others. This is because we used CE-1 fusion data and SLDEM data, and the trained designed network had a higher identification efficiency for smaller impact craters. According to Wang et al., the number of impact craters with diameters between 1 and 5 km is less than the number of identified craters. Again, the number of impact craters with larger diameters was less than that of the identified craters. At 100 km, they almost overlap, and there is also no global correction. Wang et al.'s crater center location has a different offset from the rest of the databases. Only the craters detected in CE-1 were used for comparison, which accounted for 15% of the total number of craters seen.

According to the initial study results, the accuracy of most of the craters derived from CE-1 data was D = 10~50 km. For the Sliburt et al. Impact Crater Database, the identification number was small for D ≤ 3 km and D ≥ 50 km. This indicates that compared with the deep learning method, the transfer learning-based detection identified a larger number of craters in the small and large diameter ranges with fuzzy and severe degradation. Note that it is challenging to detect the secondary craters using the automated methods.

### 4.2. Network Performance Comparison

#### 4.2.1. Comparison of Crater Detection Performance of Different Networks

We trained a total of 2 groups of 10 residual network modules in the R-FCN models, including the groups with different residual network depths of 50 and 101 layers. Using random seeds to divide data into the training set and verification set, each model operated three different sources for training. The results for each model in the validation set are shown in Table 2. The Precision, Recall, F1 Score, test time of each image, and the required memory size of the models were considered as the performance measure.

As it is seen in Table 2, for the network depth of 50 layers, the detection accuracy and recall rate increased by using various improved ResNet modules. The SCNeSt-50-FPN model achieved an accuracy rate of 89.6 and a recall rate of 81.2, which was 3% higher than that of the ResNeSt-50-FPN model. It can also be seen that adaptive convolution and different pooling methods resulted in more accurate crater contour extraction. By

increasing the depth of the network, the performance of each residual network was also improved. Compared with other residual networks, the accuracy rate and recall rate of the SCNeSt-101-FPN reached 92.7 and 90.1, respectively, and its F1 total score reached 91.3, which suggests an excellent detection result. Compared with the ResNeSt, the memory requirement of our proposed model was reduced, and the time to detect a picture was about 0.125 s.

**Table 2.** Detection index results for different networks.

| Backbone | Precision (%) | Recall (%) | $F_1$ Score (%) | Times (s) | Params (M) |
|---|---|---|---|---|---|
| ResNet-50-FPN | 79.2 | 63.5 | 70.4 | 0.140 | 25.6 |
| SCNet-50-FPN | 80.1 | 75.6 | 77.7 | 0.141 | 25.6 |
| ResNeXt-50-FPN | 84.2 | 79.3 | 81.6 | 0.132 | 25.0 |
| ResNeSt-50-FPN | 86.3 | 80.1 | 83.1 | 0.141 | 27.5 |
| SCNeSt -50-FPN | 89.6 | 81.2 | 85.2 | 0.136 | 27.5 |
| ResNet-101-FPN | 80.2 | 69.8 | 74.6 | 0.134 | 44.5 |
| SCNet-101-FPN | 82.5 | 83.2 | 82.9 | 0.135 | 44.6 |
| ResNeXt-101-FPN | 87.9 | 85.3 | 86.5 | 0.121 | 44.2 |
| ResNeSt-101-FPN | 89.3 | 88.3 | 88.7 | 0.136 | 48.2 |
| SCNeSt -101-FPN | 92.7 | 90.1 | 91.3 | 0.125 | 48.1 |

The P-R curve of the training process is shown in Figure 11. The SCNeSt model achieved the highest performance on the test dataset. This is mainly due to its improvements in pooling and the self-calibrated branch, which completed the seamless fusion of multi-scale features.

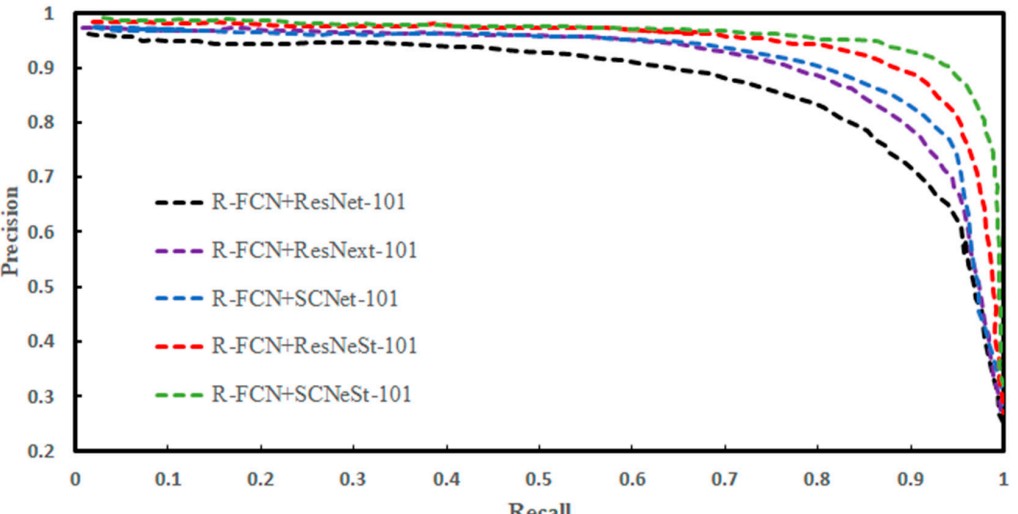

**Figure 11.** The P-R curves for different models.

To further demonstrate the results of each model, we chose 3 CE-1 fusion images and 2 SLDEM images in the verification set to compare the products, as shown in Figure 12.

Figure 12 shows samples of the impact crater detection. It is seen that the proposed model in this paper had a better detection effect on craters of different scales. Compared with the impact crater detection results of different models in Figure 12b, other models cannot detect small and prominent impact craters. It can also be seen in Figure 12c that ResNext can identify large impact craters, which is attributed to the Group Convolution. As shown in Figure 12d, some small impact craters could be accurately detected, which means that self-calibrated Conv can establish small space and inter-channel dependency around each spatial location. Therefore, it can help CNN generate feature expressions with more discriminant ability because it has more abundant information. Figure 12e also

shows that large impact craters and some minor impact craters were efficiently detected but many small impact craters were still missed. In Figure 12f, impact craters of different scales can be effectively detected. Thanks to the combination of adaptive convolution and split attention, more features can be extracted. To further test the influence of the PS-ROI Align module and Soft-NMS on the performance of the R-FCN network, two groups of control tests were conducted. The results are presented in Tables 3 and 4.

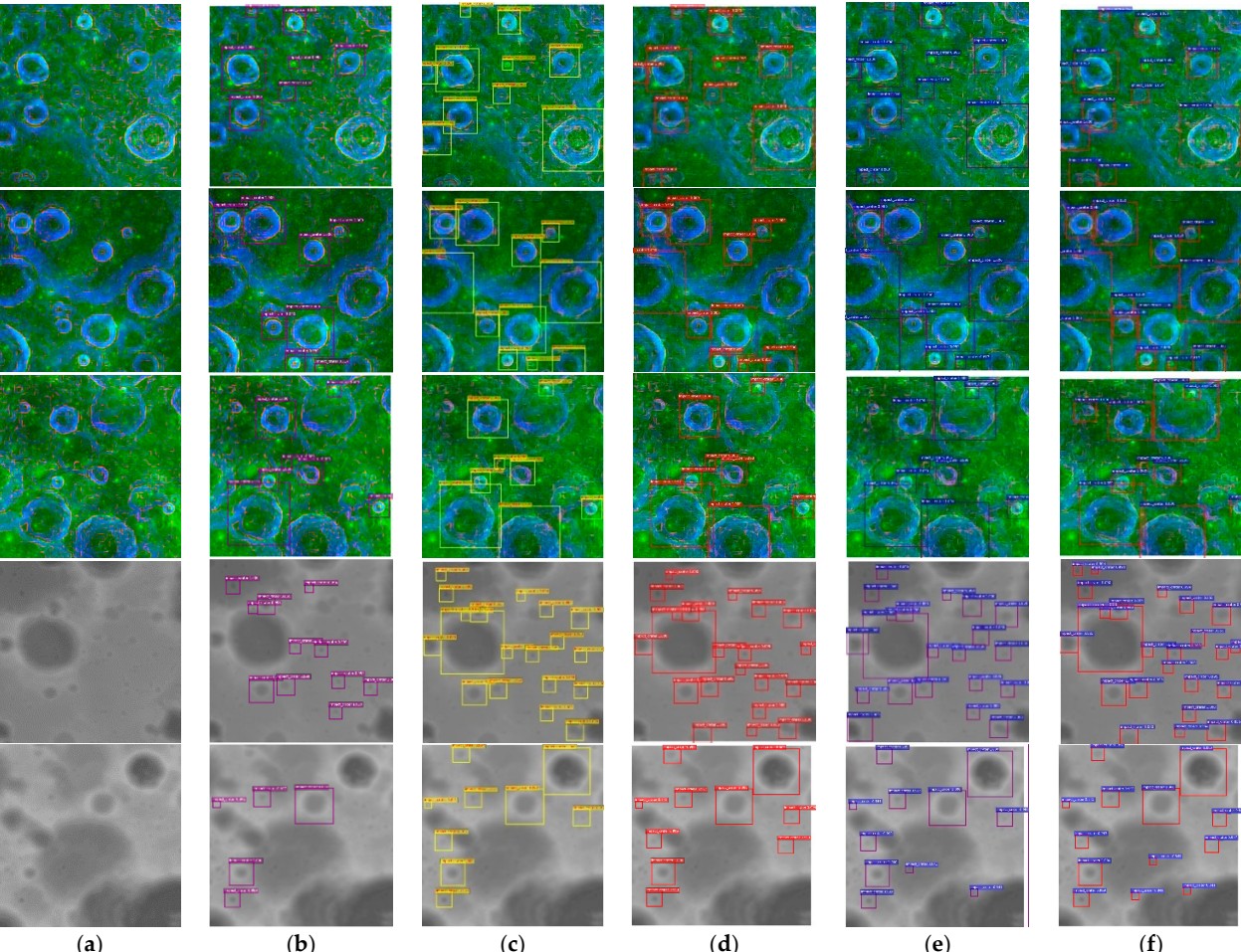

| (a) | (b) | (c) | (d) | (e) | (f) |

**Figure 12.** Comparison of the impact crater detection for different models: (**a**) Origin DEM, (**b**) ResNet, (**c**) ResNeXt, (**d**) ScNet, (**e**) ResNeSt, and (**f**) Our model.

Table 3 shows that the PS-ROI Align was superior to ROI Pooling in terms of accuracy, recall rate, and $F_1$ score at different network depths. This means that the ROI Align cancels the quantization operation. The pixels with floating-point coordinates in the quantization process were calculated by bilinear interpolation, which resulted in higher detection accuracy for small impact craters. Table 4 further shows the experimental results of the Soft-NMS and NMS detection boxes. It is seen that the improved Soft-NMS offered a higher detection performance than that of NMS. It is worth noting that the Soft-NMS needed no further training and was simple to implement. It is also simple to incorporate into any object detection operation.

### 4.2.2. Performance Comparison of Multi-Scale Impact Crater Networks

To verify the robustness and obtain the portability of the model, four lunar remote sensing data with different resolutions were selected for detection. They were SLDEM data with a resolution of 118 m/piex and 59 m/piex, LRO DEM data with a resolution of 29 m/pix, and DOM data with 7 m/pix. The test results are presented in Figure 13.

**Table 3.** Added ROI network parameter comparison.

| Basic Net | Target Detection Network | ROI Pooling | PS-ROI Align | Recall (%) | Recall (%) | $F_1$ |
|---|---|---|---|---|---|---|
| SCNeSt-50 | R-FCN | 1 | 0 | 85.3 | 79.6 | 82.3 |
| | | 0 | 1 | 86.3 | 80.1 | 83.1 |
| SCNeSt-101 | R-FCN | 1 | 0 | 90.7 | 87.1 | 88.8 |
| | | 0 | 1 | 92.7 | 90.1 | 91.3 |

**Table 4.** Added Soft-NMS network parameter comparison.

| Basic Net | Target Detection Network | NMS | Soft-NMS | Recall (%) | Recall (%) | $F_1$ |
|---|---|---|---|---|---|---|
| SCNeSt-50 | R-FCN | 1 | 0 | 85.4 | 79.6 | 80.3 |
| | | 0 | 1 | 86.3 | 80.1 | 83.1 |
| SCNeSt-101 | R-FCN | 1 | 0 | 91.2 | 88.7 | 82.9 |
| | | 0 | 1 | 92.7 | 90.1 | 91.3 |

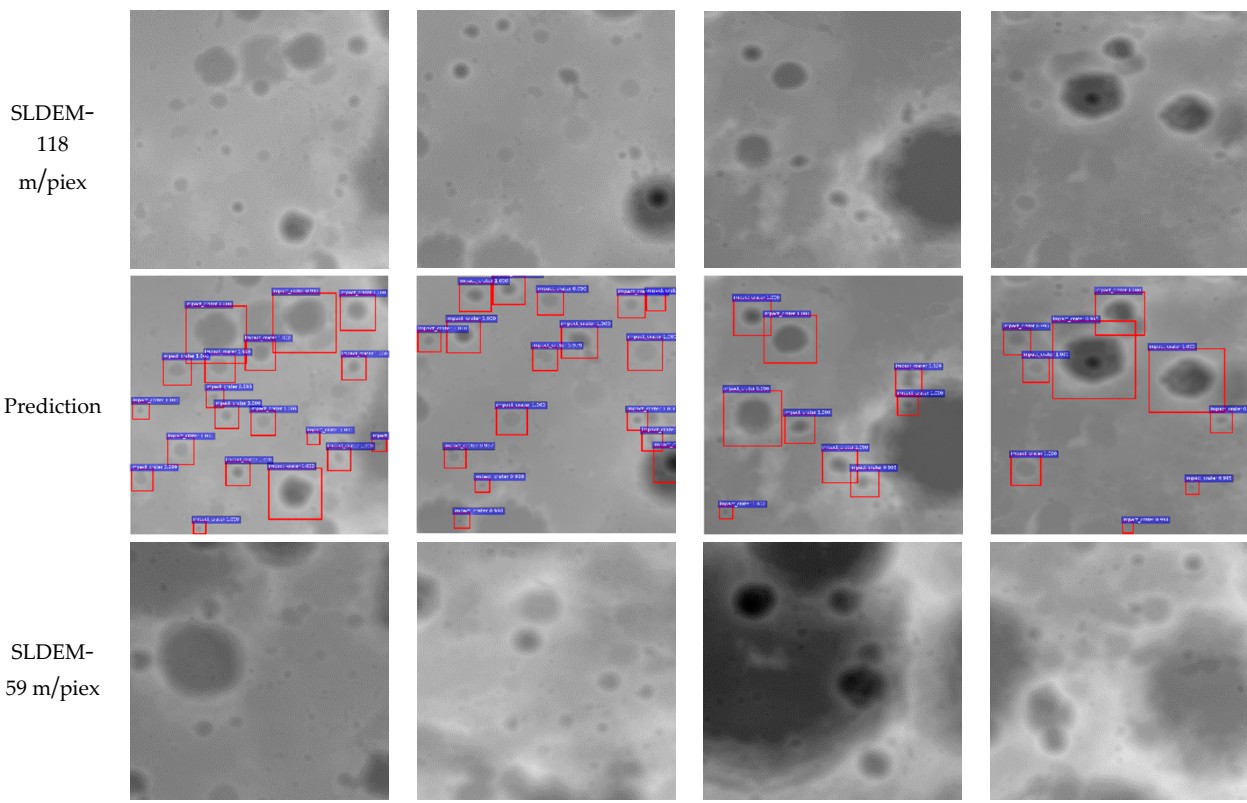

**Figure 13.** *Cont.*

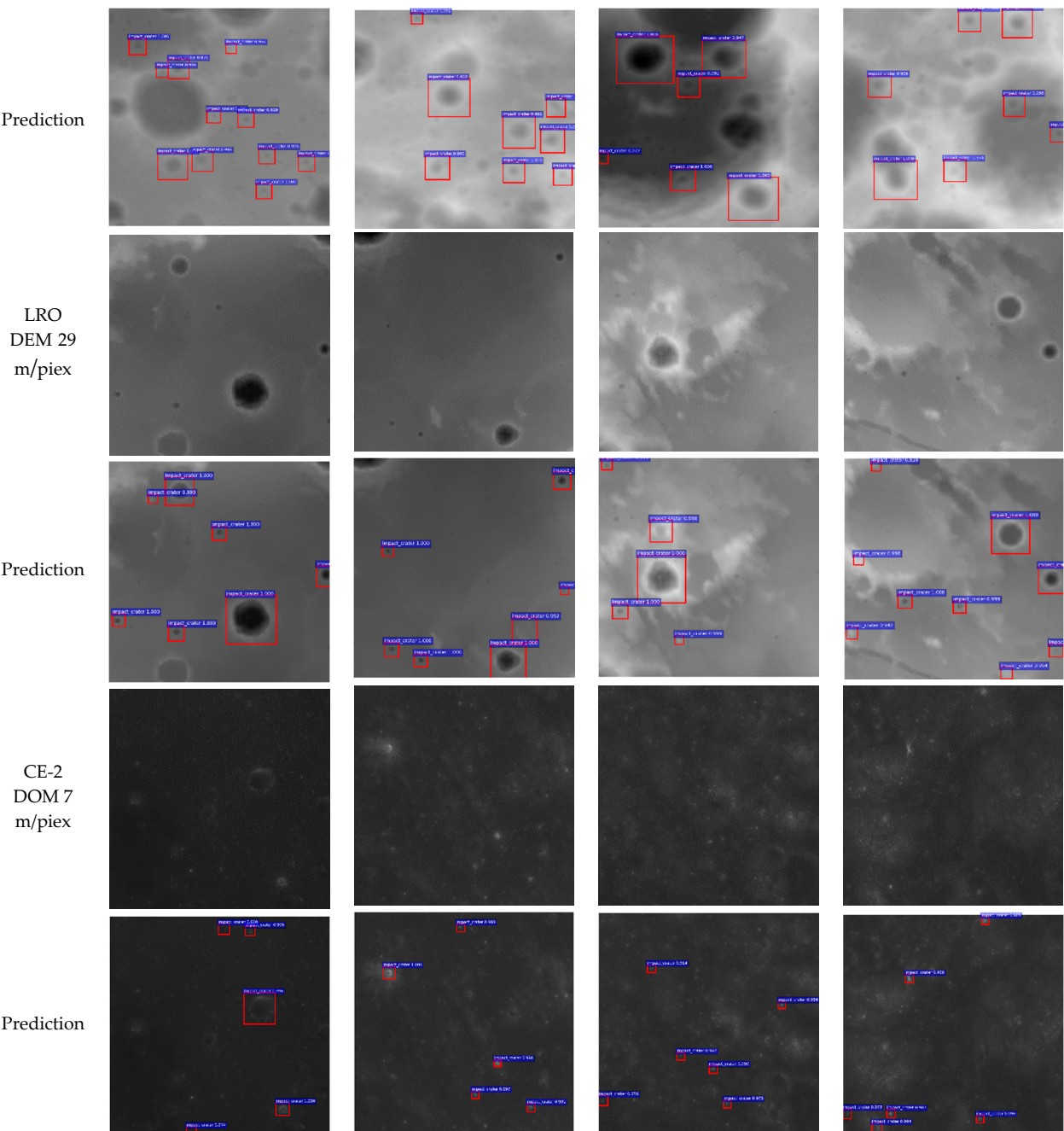

**Figure 13.** Crater detection results for data with different resolutions.

It is seen that the LRO DEM 29 m/pix results were more accurate in crater detection for different sensor resolutions. However, for more precise illumination data, the detection performance was rather low. Although some impact craters with high pixel points could be detected, most of them were not detected. This may be because DOM data is affected by illumination, which is not ideal for our model detection. For high-resolution DEM data, however, our model provided high detection performance.

### 4.3. Transfer Learning in Mars and Mercury Impact Crater Detection Analysis

Identifying the secondary impact craters is a critical step in the crater counting process for surface age determination. Failure to take these factors into account may result in a significant overestimation of the measured crater density, leading to incorrect model ages. We applied our model to Mars and Mercury data to examine the robust-

ness of our model. The MARS_HRSC_MOLA_BLENDDEM_GLOBAL_200m and MERCURY_MESSENGER_USGS_DEM_GLOBAL_665m datasets were selected for Mars and Mercury, respectively. The results are shown in Figure 14.

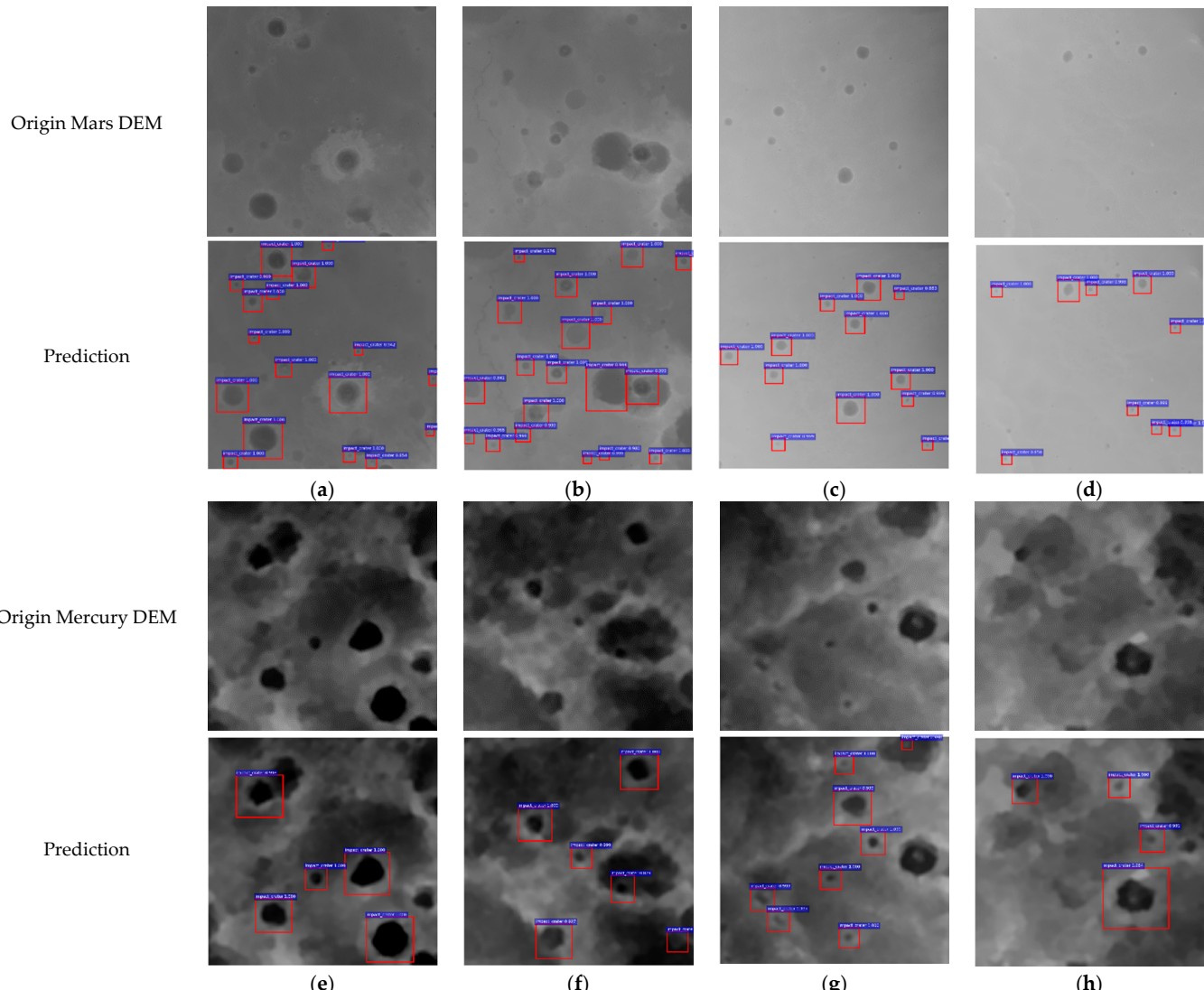

**Figure 14.** Transfer learning to Mars and Mercury impact craters. Mars crater detection results (**a**–**d**), and Mercury crater detection results (**e**–**h**).

Figure 14 shows that the detection recall rate for medium and small impact craters on Mars was 96.8, and multi-scale impact craters were detected. For Mercury, due to the resolution of the dataset and the irregular shape of the craters, some craters were miss-detected. Note that the model trained using the lunar data was applied to Mars and Mercury. In terms of the overall test results, our model achieved a high level of robustness, especially for multi-scale Mars craters.

## 5. Conclusions

In this study, a new deep-space crater detection network model was proposed, which was trained end-to-end for lunar, Mars, and Mercury data. The CE-1 DEM and DOM data were used as the training data. Based on the R-FCN network architecture, self-calibrated Conv and split attention mechanisms were used for feature extraction. Combined with the multi-scale RPN model, our proposed model efficiently extracted the features of the large, medium, and small impact craters. We further introduced a Position-Sensitive

ROI Align network structure that can effectively remove the contour of irregular impact craters. Combined with the improved Soft-NMS framework, the overlapping craters can be efficiently detected. Our model evaluated the proposed network on four resolution lunar data and Mars and Mercury data through transfer learning, and the results demonstrated its advantages for crater-detection missions. Therefore, we will continue to look for small impact craters (D < 1 km) to lay the groundwork for lunar and Mars lander landings and navigation applications.

**Author Contributions:** Data curation, R.Y.; Funding acquisition, G.W.; Project administration, Y.W. (Yitian Wu); Resources, J.W.; Software, L.L.; Validation, Y.W. (Ying Wang); Visualization, N.X.; Writing—original draft, Y.J.; Writing—review & editing, Y.J. All authors have read and agreed to the published version of the manuscript.

**Funding:** This research received no external funding.

**Data Availability Statement:** In this study, using Chang-E data download address for Chinese lunar exploration data and information system, web site for https://moon.bao.ac.cn/moonGisMap.search (accessed on 4 July 2021). In addition, the use of the LRO DEM data and SLDEM data, as well as Mars and Mercury in the USGS DEM data, download website, https://planetarymaps.usgs.gov/mosaic/ (accessed on 4 July 2021). International Astronomical Union. https://planetarynames.wr.usgs.gov/Page/MOON/target (accessed on 4 July 2021).

**Acknowledgments:** The authors would like to thank Space Engineering University for its hardware support and NASA's Lunar digital elevation model data. In addition, the author is incredibly grateful to Zhao Haishi for his advice.

**Conflicts of Interest:** The authors declare no conflict of interest.

## Abbreviations

The following abbreviations are used in this manuscript:

| | |
|---|---|
| CDA | Crater detection algorithm |
| LRO | Lunar Reconnaissance Orbiter |
| MOLA | Mars Orbiter Laser Altimeter |
| MOC | Mars Orbiter Camera |
| HRSC | High Resolution Stereo Camera |
| CNN | Convolutional neural networks |
| IAU | International Astronomical Union |
| RPN | Region proposal network |
| NMS | Non-maximum suppression |
| RoI | Region of interest |
| FPN | Feature pyramid network |
| DEM | Digital Elevation Model |
| DTM | Digital Terrestrial Model |
| DOM | Digital Orthophoto Map |

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
