# Peer review of "Split-Attention Networks with Self-Calibrated Convolution for Moon Impact Crater Detection from Multi-Source Data"

_remotesensing, doi:10.3390/rs13163193_

Round 1

Reviewer 1 Report

The paper presents a system to identify craters from images of Mercury, Mars and the Moon using deep neural networks. The proposed model (split attention network architecture with self-calibrating convolution (SCNeSt)) demonstrated better performance than other approaches for detecting craters of different dimensions.   The study was well conducted, however, some issues need to be addressed:   1. The description of the database can be improved. Explain characteristics of different craters with respect to different areas and sizes and their implications on the results. 2. Some references in the figure are erroneous, such as.  Figure 3 - page 3 "image merging method referred to 3.1" - page 3 3. It is difficult to understand the meaning of different acronyms, please add the meaning of all acronyms.   4. I am unable to identify what percentage of the images were used for training and testing and what type of validation was used.   5. I would like the explanation of the feature extraction process to be explained in a more compact way and in general the whole methodology in a paragraph at the beginning of the experiments section.

Author Response

Point 1: The description of the database can be improved. Explain characteristics of different craters with respect to different areas and sizes and their implications on the results.

Response 1: On page 10, we explain CE-1 data and SLDEM data and explain the identification of different impact craters with different data.

Point 2: Response to comment:  Some references in the figure are erroneous, such as.  Figure 3 - page 3 "image merging method referred to 3.1" - page 3.

Response 2: We are very sorry for our incorrect writing. The correct one should be as shown in Figure 1. We have modified it on page 3.

Point 3: Response to comment: It is difficult to understand the meaning of different acronyms, please add the meaning of all acronyms.

Response 3: We add abbreviations on Page 19 to express different acronyms meaning.

Point 4: I am unable to identify what percentage of the images were used for training and testing and what type of validation was used.

Response 4: ‘The CE-1 fusion data is then clipped into 1200×1200, 4800×4800 images at a 50% overlap rate, 8000,1000 and 1000 images are randomly selected and used for training, validation, and testing, respectively.’  We underline in yellow on page 10.

Point 5. Response to comment: I would like the explanation of the feature extraction process to be explained in a more compact way and in general the whole methodology in a paragraph at the beginning of the experiments section.

Response 5: It is really true as Reviewer suggested that we have a supplementary explanation of the experiment on line 275 on page 9. Our algorithm is divided into two parts. Firstly, the features of impact craters are ex-tracted under the Structure of the R-FCN network based on the SCNeSt network skeleton, and the data is DOM and DEM fusion data from CE-1. Multi-scale Feature Extractor and Position-Sensitive ROI Align can better detect impact craters of different scales. Combined with Soft-NMS algorithm to accurately convey the best coordinates of the target and remove the repeated boundary box. In the first stage, the craters with D>20km were mainly ex-tracted. In the second stage, the trained model was applied to SLDEM data to extract small craters with D<20km. What's more, The trained models are then applied to detecting the impact craters on Mercury and Mars using the transfer learning method.

Reviewer 2 Report

The manuscript " Split-Attention Networks with Self-Calibrated Convolution for Moon 
Impact Craters Detection from Multi-Source Data " (remotesensing-1307236), proposed a model to efficiently extract the features of large, medium, and small craters. The study is clear and the results are also comprehensive. Besides, this paper is well-organized and easy to read. There are some flaws:  

1) Some figures are not referenced in the text (e.g. Fig.1 on page 4 and Fig.8 on page 10 - in fact there are two figures with the number 8, page 10 and page 12) and some figures appear before they are referenced in the text (e.g. Fig. 3). All figures and respective references should be reviewed. Greater care should have been taken because, as it is, it created great difficulty in reviewing the article. 

Author Response

Point 1: Some figures are not referenced in the text (e.g. Fig.1 on page 4 and Fig.8 on page 10 - in fact there are two figures with the number 8, page 10 and page 1. 

Response 1: We are very sorry for our incorrect writing. We have corrected all the numbers of the pictures after page 12 and revised the quotation format.

Point 2: Some figures appear before they are referenced in the text (e.g. Fig. All figures and respective references should be reviewed. Greater care should have been taken because, as it is, it created great difficulty in reviewing the article.

Response 2: We are very sorry for the mistake. We have changed the order on the sixth page and carefully checked the quoting situation of the whole text. The modification has been completed.
